# Death of a parent during childhood and blood pressure in youth: a population-based cohort study of Swedish men

Hua Chen [1], Tomas Hemmingsson,[2,3] Imre Janszky,[1,4] Mikael Rostila,[2,5] Yvonne Forsell,[1,6] Linghui Meng,[1,7] Yajun Liang,[1] Krisztina D. László [1]

For numbered affiliations see end of article.

**Correspondence to**
Dr Hua Chen; hua.chen@ki.se

## ABSTRACT

**Objective** Compelling evidence suggests that childhood adversities are associated with an increased risk of hypertension in middle age and old age. The link between childhood adversities and blood pressure in youth is less clear. In this cohort study, we examined the association between death of a parent during childhood and blood pressure in early adulthood in men.

**Setting** Sweden.

**Participants** We studied 48 624 men born in 1949–1951 who participated in the compulsory military conscription in 1969/1970 in Sweden. Information on death of a parent during childhood was obtained from population-based registers. Information on covariates was obtained from the questionnaire and the clinical examination completed at conscription and from population-based registers.

**Outcome measures** Blood pressure was measured at conscription according to standard procedures.

**Results** The multivariable least square means of systolic and diastolic blood pressure did not differ between bereaved (128.25 (127.04–129.46) and 73.86 (72.89–74.84) mm Hg) and non-bereaved study participants (128.02 (126.86–129.18) and 73.99 (73.06–74.93) mm Hg). Results were similar when considering the cause of the parent's death, the gender of the deceased parent or the child's age at loss. Loss of a parent in childhood tended to be associated with an increased hypertension risk (OR and 95% CI: 1.10 (1 to 1.20)); the association was present only in case of natural deaths.

**Conclusion** We found no strong support for the hypothesis that stress following the loss of a parent during childhood is associated with blood pressure or hypertension in youth in men.

## INTRODUCTION

Death of a parent is one of the most traumatic and stressful events that a child can experience.[1] In the Nordic countries around 4% of children experience parental death before the age of 18 years,[2,3] while in the developing countries the prevalence is higher.[4,5] Children exposed to parental death have an increased risk of experiencing poor social support,[6] heightened stress reactivity,[7] poor school performance,[8] emotional and behavioural problems,[6] psychiatric

### Strengths and limitations of this study

► This is the first study to investigate prospectively the association between parental death in childhood and blood pressure measured in early adulthood in a large national cohort; the prospective design limited recall bias, while the large national sample allowed to reduce selection bias and to conduct subanalyses to test hypotheses that may be supportive of a causal effect, if any.

► Since only one or two blood pressure measurements were taken, there was no possibility to consider intra-individual variations in blood pressure or to reduce the impact of white-coat hypertension.

► Information on some potential confounders was not available.

► It is not clear to what extent our findings are generalisable to women, to current Swedish young adults or to populations with different risk factor profiles, support for bereaved children and healthcare.

disorders,[9,10] substance use,[11,12] obesity[13] and/or metabolic syndrome.[14] These in turn may induce stress-related physiological changes related to the dysregulation of the hypothalamic–pituitary–adrenocortical axis and of the sympathetic nervous system; these enhance pro-inflammatory activity and may overactivate the renin–angiotensin–aldosterone system.[15] Chronic stress may also lead to unhealthy lifestyle.[16] Such physiological and behavioural changes may increase blood pressure (BP) and the risk of other cardiovascular diseases (CVD) (eg, ischaemic heart diseases, stroke) or cardiac mortality,[17,18] already in young age.[2]

To our knowledge, only five studies investigated the association between death of a parent and BP and the findings have been inconsistent.[4,7,19–21] Luecken[7] and Anderson *et al*[20] reported that study participants who experienced parental death during childhood had higher systolic BP (SBP) and diastolic BP (DBP) than their unexposed counterparts. In contrast, Schooling *et al*[4]

and Stein *et al*[21] found no association between parental death and BP or hypertension. In a study conducted later, Luecken *et al* found that university students who lost their parent had lower 24-hour ambulatory BP compared with non-bereaved students.[19] Except for the studies conducted by Luecken *et al*, which assessed BP in childhood or in young adulthood,[7 19] all other studies were conducted in middle-aged individuals and assessed exposure retrospectively[4 20 21]; thus they could not eliminate selection bias, that is, that patients with severe hypertension may die before middle-age. A further potential explanation for the discrepant findings in these earlier studies may be related to differences in their multivariate adjustments; some investigators did not adjust at all for confounders,[7 19] some adjusted for a limited number of confounders,[4 21] while others adjusted for several factors that may be on the causal pathway between parental death and hypertension.[20]

None of these earlier studies investigated the importance of the type of the parent's death, analyses often conducted in bereavement research to attempt to separate the effect of stress from confounding or to investigate dose-response effects. The effect of exposure to parental death due to natural causes, in particular those due to CVD, on the risk of hypertension is likely to be confounded by genetic and environmental cardiovascular factors shared by family members, for example, socioeconomic factors, lifestyle, mental health, cardiometabolic risk factors and morbidity; in contrast, the importance of familial confounding by cardiovascular risk factors is likely to be substantially more modest for unnatural deaths.[2 22] Furthermore, unnatural deaths have been suggested to be more strongly associated with stress and an increased risk of complicated grief than deaths due to natural causes.[6 16 23] Similarly, maternal death may have a more negative effect on BP than paternal death since in most cultural contexts mothers have stronger emotional bonds with their children, are more involved in their children's upbringing,[24] have a stronger impact on their children's health behaviour and are likely to provide more emotional support for coping with grief than fathers.[25] Similarly, the first few years of life and adolescence could be particularly sensitive periods with respect to stress. Losing a parent in the first few years of life may be particularly detrimental[26] as warm relationships with caregivers are critical for the development of the brain architecture and the programming of stress reactivity,[7] while loss of a parent in adolescence, another period with an increased stress sensitivity, may result in taking up adverse health behaviours that increase the CVD risk.[27]

High BP in youth tracks into adulthood and is an important predictor of later hypertension and CVD[28 29]; it is associated with increased risks of abnormal left ventricular mass,[30] metabolic syndrome,[31] impaired cognitive ability,[32] renal damage[33] and cardiovascular and total mortality.[34 35] The prevalence of elevated BP among children and young adults has been increasing.[36 37] Though adverse childhood experiences (most often defined in terms of maltreatment and family dysfunction), are associated with an increased risk of hypertension in middle age and old age,[15 38 39] knowledge about their association with high BP in youth is more limited.[40–43]

We investigated the association between parental death during childhood—one of the most severe childhood adversities—and BP at the age of 18–20 years in a cohort of Swedish men. We also analysed whether this association differs by the parent's gender, cause of death and the child's age at loss.

## METHODS
### Study population and design
This study was based on a cohort consisting of 49 321 men born in 1949–1951 and who were conscripted for military service in Sweden in 1969–1970. The cohort includes 97%–98% of all Swedish men of conscription age (≈18–20 years); the 2%–3% that were exempted from participation had severe congenital disorders or intellectual disabilities.[44] At conscription, participants completed an extensive questionnaire and participated in a clinical examination performed by a team of physicians, psychiatrists and psychologists. The conscription data were linked to several nationwide registers through the unique personal number. We excluded participants with no information on exposure (n=314), missing BP (n=369) or higher DBP than SBP (n=14), resulting in 48 624 men being included in our analysis.

### Measures
#### Exposure
Conscripts were linked to their parents using the Multigeneration Register. We obtained information on the date and the cause of the parent's death after 1952 from the Cause of Death Register; causes of death were recorded according to the International Classification of Diseases (ICD). We classified the cause of the parent's death as due to unnatural causes (ICD-6: 795 and 800–999; ICD-7: 795 and 800–999; ICD-8: 79599, 796 and 807–999), due to non-rheumatic CVD (ICD-6: 400–468; ICD-7: 400–468; ICD-8: 400–458) and due to other natural causes (the rest of the ICD codes). For the conscripts whose parents did not appear in the Cause of Death Register, we used two items from the questionnaire completed at conscription to determine their exposure status before 1952: 'Are both of your parents alive' (yes/no), and 'Whom have you mostly lived with' (with both parents/mother/father/somebody else). If study participants responded 'no' to the first question, the gender of the deceased parent was determined using the second question: (1) having lived with the mother was coded as paternal death, (2) having lived with the father was coded as maternal death and (3) having lived with both parents or somebody else was coded as having no information on the deceased parent's gender.

#### Outcome
BP was measured at conscription according to a written protocol; the measurement was performed in a supine

position, after a 5–10 min rest, with an appropriately sized cuff at the heart level.[44–46] Thus the measurement of BP in this study reflects the standard practice in primary care settings.[46] The measured SBP and the DBP values were rounded to the nearest even number, or to the nearest 5 or 10 mm Hg. If the SBP was higher than 145 mm Hg or if the DBP was lower than 50 mm Hg or higher than 85 mm Hg, another measurement was conducted the next day and the result of the second measurement was recorded; otherwise only one BP measurement was taken. Given that there is a linear association between BP in young age and the later risk of CVD,[47] that even BP values below 140/90 mm Hg have a predictive value for the development of later CVD, and since both SBP and DBP are associated with later CVD risk,[48 49] we considered the continuous SBP and DBP measures as the main outcome. In sensitivity analyses we also considered hypertension, defined as SBP ≥140 mm Hg and/or DBP ≥90 mm Hg.[50]

## Covariates

The occupation of the head of the household (generally the father) from the National Population and Housing Censuses of 1960 was used as a measure of parental socioeconomic status (SES). Occupation was classified as: non-manual at middle or high level, non-manual employee at low level, skilled worker, unskilled worker, farmer and other.[51] Men who during the psychological interview reported or were suspected to have a psychiatric disorder were referred to a psychiatrist for further evaluation. We retrieved information on depression using ICD-8 codes 296 and 300.4 and on anxiety using code 300.0, respectively. We calculated body mass index based on height and weight assessed at the clinical examination. Cardiorespiratory fitness was assessed using a cycle ergometer submaximal exercise test performed after a normal resting ECG. The work rate was increased until the participants were volitionally exhausted; the resulting maximal work capacity divided by weight was transformed to stanine scores (1–9).[45] The number of personal friends (0, 1–3, 3–5 or >5), the frequency of confidentially discussions with friends (never, sometimes or quite often), having a confidant to talk about personal problems (no one, or parent/siblings/teachers or supervisor or manager/friends/others), number of cigarettes per day (>10, 1–10 or 0), risky drinking behaviour (defined as reporting any of the following: drinking >250 g/week, having taken alcohol as an 'eye-opener' during a hang-over, having been arrested for drunkenness or having often been intoxicated)[52] and father's drinking habit (often, or sometimes/occasionally/never) were assessed by questionnaires. We calculated a cumulative index of childhood adversities by summing the number of the adverse childhood experiences reported by questionnaire from the following: financial situation of the family (bad/very bad vs very good/good/average), paternal alcohol use (often vs never/occasionally/sometimes), parents' divorce (yes vs no), severe illness of parent (father/mother/both parents vs none), family member

taking medicines for mental disorders (mother/father/both parents/others vs no one), experiencing physical punishment (often/sometimes/once in a while vs never), strict upbringing (very strict/quite strict vs medium/quite mild/very mild) and having multiple residences during childhood (more than three residences vs one/two/three residences). We considered participants whose answer was in the first group to have been exposed to the corresponding adverse experience. We chose these adverse childhood experiences based on previous literature—which often defined adverse childhood experiences in terms of abuse (emotional, physical and sexual), neglect or family dysfunction[15 53 54]—and the questionnaire completed at conscription. We calculated the cumulative index by summing the number of adverse events experienced, a common praxis in the literature[15 53 55] and based on evidence of a dose-response association between the number of adverse childhood experiences and the risk of CVD.[53 55]

## Statistical analyses

We compared characteristics of exposed and unexposed men by $\chi^2$ tests in case of categorical variables and Wilcoxon tests in case of continuous variables with a skewed distribution. We used similar tests and logistic regression to study the link between covariates and our outcomes. We performed general linear regression to investigate the association between parental death and the continuous SBP and DBP measures. For each exposure category, we also estimated least square means of SBP and DBP and their 95% CIs. We performed analyses with any parental death during childhood and with exposure classified based on the cause of the parent's death, the gender of the deceased parent and the age of the child when the parent died. We ran three models. Model 1 was unadjusted. Model 2 was adjusted for childhood parental SES, a factor that is likely to be confounder of the investigated association. Model 3 was further adjusted for depression, anxiety, body mass index, cardiorespiratory fitness, number of friends, frequency of talking with friends confidentially, having a confidant, number of cigarettes smoked per day and risky drinking behaviour. We chose to run model 3 separately since we could not determine whether the variables in model 3 are confounders or mediators of the association under study; though their assessment after the exposure period may favour regarding them as mediators, several of these characteristics could have been present before the loss of a parent and may thus be confounders. To study whether father's drinking behaviour confounded the association between death of a father and BP, we re-ran these analyses after adjusting for father's drinking behaviour in addition to factors in model 3. To assure that the method of exposure assessment did not influence our results, we ran analyses after excluding the conscripts whose parent's death was identified only by questionnaire. We investigated effect modification by parental SES (classified as (1) non-manual employee versus (2) unskilled workers or skilled

**Table 1** Characteristics of study participants according to death of a parent (N=48 624)

| | | Death of a parent | | |
| | | Yes | No | |
| Variables | Total, n (%) | (n=3504) | (n=45 120) | P value* |
|---|---|---|---|---|
| **Categorical variables, n (%)** | | | | |
| Parental socioeconomic status | | | | <0.01 |
| Middle or high level non-manual employee | 10 716 (22) | 689 (19.7) | 10 027 (22.2) | |
| Low level non-manual employee | 4942 (10.2) | 300 (8.6) | 4642 (10.3) | |
| Skilled worker | 10 403 (21.4) | 510 (14.6) | 9893 (21.9) | |
| Unskilled worker | 16 094 (33.1) | 1121 (32) | 14 973 (33.2) | |
| Farmer | 5373 (11) | 399 (11.4) | 4974 (11) | |
| Other | 1096 (2.3) | 485 (13.8) | 611 (1.4) | |
| Depression | | | | <0.01 |
| No | 47 892 (98.5) | 3421 (97.6) | 44 471 (98.6) | |
| Yes | 732 (1.5) | 83 (2.4) | 649 (1.4) | |
| Anxiety | | | | 0.72 |
| No | 48 469 (99.7) | 3494 (99.7) | 44 975 (99.7) | |
| Yes | 155 (0.3) | 10 (0.3) | 145 (0.3) | |
| Cardiorespiratory fitness (stanine scores) | | | | <0.01 |
| 1 | 34 (0.1) | 2 (0.1) | 32 (0.1) | |
| 2 | 207 (0.4) | 15 (0.4) | 192 (0.4) | |
| 3 | 2451 (5) | 207 (5.9) | 2244 (5) | |
| 4 | 7020 (14.4) | 578 (16.5) | 6442 (14.3) | |
| 5 | 11 447 (23.5) | 857 (24.4) | 10 590 (23.5) | |
| 6 | 9215 (19) | 658 (18.8) | 8557 (19) | |
| 7 | 5713 (11.8) | 401 (11.4) | 5312 (11.8) | |
| 8 | 4608 (9.5) | 305 (8.7) | 4303 (9.5) | |
| 9 | 7867 (16.2) | 480 (13.7) | 7387 (16.4) | |
| Missing | 62 (0.1) | 1 (0.1) | 61 (0.1) | |
| Number of personal friends | | | | 0.72 |
| 0 | 389 (0.8) | 29 (0.8) | 360 (0.8) | |
| 1–3 | 3260 (6.7) | 240 (6.8) | 3020 (6.7) | |
| 3–5 | 15 195 (31.3) | 1114 (31.8) | 14 081 (31.2) | |
| >5 | 29 006 (59.6) | 2048 (58.4) | 26 958 (59.7) | |
| Missing | 774 (1.6) | 73 (2.1) | 701 (1.6) | |
| Confidential discussions with friends | | | | <0.01 |
| Never | 2011 (4.2) | 178 (5.1) | 1833 (4.1) | |
| Sometime | 28 655 (58.9) | 2061 (58.8) | 26 594 (58.9) | |
| Quite often | 17 072 (35.1) | 1186 (33.8) | 15 886 (35.2) | |
| Missing | 886 (1.8) | 79 (2.3) | 807 (1.8) | |
| Has a confidant | | | | <0.01 |
| No | 7865 (16.2) | 633 (18.1) | 7232 (16) | |
| Yes | 39 741 (81.7) | 2772 (79.1) | 36 969 (81.9) | |
| Missing | 1018 (2.1) | 99 (2.8) | 919 (2) | |
| Number of cigarettes smoked per day | | | | |
| 0 | 19 903 (40.9) | 1216 (34.7) | 18 687 (41.4) | <0.01 |
| 1–10 | 15 310 (31.5) | 1150 (32.8) | 14 160 (31.4) | |

Continued

| | | **Death of a parent** | | |
| | | **Yes** | **No** | |
| **Variables** | **Total, n (%)** | **(n=3504)** | **(n=45 120)** | **P value*** |
| >10 | 12 731 (26.2) | 1067 (30.5) | 11 664 (25.9) | |
| Missing | 680 (1.4) | 71 (2) | 609 (1.3) | |
| Risky drinking behaviour | | | | <0.01 |
| No | 42 323 (87) | 2923 (83.4) | 39 400 (87.3) | |
| Yes | 6267 (12.9) | 579 (16.5) | 5688 (12.6) | |
| Missing | 34 (0.1) | 2 (0.1) | 32 (0.1) | |
| Father's drinking habits | | | | <0.01 |
| Never, occasionally or sometimes | 45 580 (93.8) | 3048 (87) | 42 532 (94.3) | |
| Often | 1954 (4) | 175 (5) | 1779 (3.9) | |
| Missing | 1090 (2.2) | 281 (8) | 809 (1.8) | |
| Cumulative adverse childhood experiences | | | | <0.01 |
| 0 | 9617 (19.8) | 482 (13.8) | 9135 (20.2) | |
| 1 | 18 477 (38) | 1043 (29.8) | 17 434 (38.6) | |
| 2 | 12 173 (25) | 1061 (30.3) | 11 112 (24.6) | |
| ≥3 | 8357 (17.2) | 918 (26.2) | 7439 (16.5) | |
| **Continuous variable, median** | | | | |
| Body mass index, kg/m$^2$ | 20.6 | 20.6 | 20.7 | 0.71 |

*Men with no missing data on the corresponding variables were included in these analyses.

workers versus (3) farmer or other) and by our cumulative index of adverse childhood experiences (0, 1, 2 or ≥3) by stratified analyses and formal tests of interaction. In further sensitivity analyses we run logistic regression models to investigate the association between death of a parent (any loss and exposure classified by the parent's cause of death, the deceased parent's gender and the child's age at loss) and hypertension. We deleted listwise in case of missing information on covariates.

We used SAS 9.4 for Windows for the analyses.

### Patient and public involvement

Patients and/or the public were not involved in the design, conduct, reporting or dissemination plans of this research.

### RESULTS

Of the 48 624 men included in our study, 3504 (7.21%) experienced death of a parent before the age of 18. Characteristics of exposed and unexposed participants are shown in table 1. The association between covariates and the risk of our outcomes is presented in online supplemental table 1.

The least square means of SBP and DBP were generally similar between the exposed and the unexposed groups, both when investigating any loss of a parent and when the exposure was classified according to the deceased parent's cause of death and gender, or the child's age at loss (table 2). Men whose mother died due to unnatural causes had a slightly lower DBP than the unexposed in the unadjusted and the SES-adjusted, but not in the fully adjusted models (table 3). The results did not change after excluding conscripts whose parent's death was defined through questionnaire-based information (online supplemental table 2). We found no strong evidence that parental SES or cumulative childhood adversity modified the association between parental death and BP (online supplemental table 3). Adjusting for father's drinking behaviour in addition to factors in model 3 did not substantially change the association between death of a father and SBP or DBP (data not shown).

Death of a parent tended to be associated with a slightly increased odds of hypertension; the corresponding multivariate OR (95% CI) was 1.10 (1 to 1.20). Losing a parent due to natural causes other than CVD was associated with hypertension. Losing a parent due to CVD also tended to be associated with hypertension. There was no association between parental death due to unnatural causes and the risk of hypertension. The point estimates corresponding to the association between death of a parent and hypertension did not differ substantially according to the parent's gender. Risks were slightly higher in case the child was 6–12 or 13–18 years at loss compared with when the loss occurred at earlier ages; these associations were confined only to losses due to natural causes (table 4).

**Table 2** Least square means and 95% CIs for blood pressure according to exposure to death of a parent during childhood (N=48 624)

| Type of exposure | Model 1* | | Model 2† | | Model 3‡ | |
|---|---|---|---|---|---|---|
| | LS mean (95% CI) | P value | LS mean (95% CI) | P value | LS mean (95% CI) | P value |
| **Systolic blood pressure, mm Hg** | | | | | | |
| Unexposed | 126.09 (125.99 to 126.20) | – | 126.13 (125.97 to 126.29) | – | 128.02 (126.86 to 129.18) | – |
| Any loss | 126.34 (125.95 to 126.72) | 0.24 | 126.35 (125.96 to 126.74) | 0.30 | 128.25 (127.04 to 129.46) | 0.26 |
| Cause of death of the parent§ | | | | | | |
| Unnatural death | 125.49 (124.58 to 126.40) | 0.20 | 125.56 (124.65 to 126.47) | 0.23 | 127.57 (126.10 to 129.03) | 0.59 |
| Cardiovascular death | 126.94 (126.01 to 127.87) | 0.08 | 126.99 (126.06 to 127.92) | 0.07 | 128.40 (126.92 to 129.88) | 0.22 |
| Other natural cause | 126.62 (126.05 to 127.19) | 0.08 | 126.59 (126.02 to 127.16) | 0.12 | 128.31 (127.02 to 129.59) | 0.10 |
| Gender of deceased parent§ | | | | | | |
| Mother | 126.36 (125.59 to 127.14) | 0.50 | 126.26 (125.47 to 127.04) | 0.74 | 128.14 (126.74 to 129.53) | 0.51 |
| Father | 126.50 (126 to 126.99) | 0.12 | 126.55 (126.06 to 127.05) | 0.10 | 128.30 (127.04 to 129.55) | 0.11 |
| Child's age at loss | | | | | | |
| ≤5 years | 125.93 (125.24 to 126.62) | 0.65 | 125.96 (125.26 to 126.65) | 0.60 | 127.87 (126.54 to 129.20) | 0.63 |
| 6–12 years | 126.30 (125.57 to 127.04) | 0.58 | 126.30 (125.56 to 127.03) | 0.69 | 128.15 (126.79 to 129.51) | 0.77 |
| 13–18 years | 126.67 (126.06 to 127.28) | 0.07 | 126.69 (126.08 to 127.31) | 0.08 | 128.64 (127.34 to 129.94) | 0.05 |
| **Diastolic blood pressure, mm Hg** | | | | | | |
| Unexposed | 72.90 (72.82 to 72.99) | – | 72.94 (72.81 to 73.07) | – | 73.99 (73.06 to 74.93) | – |
| Any loss | 72.88 (72.57 to 73.19) | 0.87 | 72.86 (72.55 to 73.17) | 0.63 | 73.86 (72.89 to 74.84) | 0.45 |
| Cause of death of the parent§ | | | | | | |
| Unnatural death | 72.16 (71.43 to 72.88) | 0.05 | 72.10 (71.37 to 72.82) | 0.02 | 73.20 (72.02 to 74.39) | 0.06 |
| Cardiovascular death | 73.42 (72.68 to 74.16) | 0.18 | 73.40 (72.66 to 74.14) | 0.24 | 74.14 (72.94 to 75.33) | 0.60 |
| Other natural cause | 73.13 (72.67 to 73.58) | 0.34 | 73.12 (72.67 to 73.58) | 0.46 | 74.14 (73.10 to 75.18) | 0.40 |
| Gender of deceased parent§ | | | | | | |
| Mother | 72.36 (71.74 to 72.97) | 0.08 | 72.39 (71.76 to 73.01) | 0.09 | 73.43 (72.31 to 74.56) | 0.16 |
| Father | 73.16 (72.77 to 73.55) | 0.20 | 73.11 (72.72 to 73.51) | 0.38 | 74.00 (72.99 to 75.01) | 0.58 |
| Child's age at loss | | | | | | |
| ≤5 years | 72.62 (72.07 to 73.17) | 0.32 | 72.57 (72.02 to 73.12) | 0.20 | 73.49 (72.42 to 74.57) | 0.08 |
| 6–12 years | 73.04 (72.46 to 73.62) | 0.66 | 72.99 (72.41 to 73.57) | 0.88 | 73.98 (72.88 to 75.08) | 0.93 |
| 13–18 years | 72.97 (72.49 to 73.45) | 0.80 | 72.99 (72.50 to 73.49) | 0.84 | 74.09 (73.04 to 75.14) | 0.74 |

*Model 1 was unadjusted.
†Model 2 was adjusted for parental socioeconomic status.
‡Model 3 was adjusted for parental socioeconomic status, depression, anxiety, body mass index, cardiorespiratory fitness, number of friends, frequency of talking with friends confidentially, having a confidant, number of cigarettes smoked per day and risky drinking behaviour.
§Only men with no missing data on this type of exposure were included.
LS, least square.

## DISCUSSION

We found no strong evidence for an association between the death of a parent during childhood and SBP or DBP in early adulthood. The associations did not differ by the parent's cause of death, the gender of the deceased parent or the child's age at loss. Losing a parent was associated with a slightly increased risk of hypertension; the association was present only in case of losses due to natural causes.

Earlier studies investigating the association between parental death during childhood and elevated BP or hypertension have yielded mixed results. Two studies observed higher SBP and DBP among participants exposed to parental death during childhood than among their unexposed counterparts.[7 20] Two other studies found no relation between parental death and BP or hypertension,[4 21] whereas one study observed that students who experienced parental death had lower 24-hour ambulatory BP than non-bereaved students.[19] The reasons for these inconsistent findings are not clear but could be due to differences in study design, sample size, historical birth cohort, the age and the method of BP assessment and considerations about confounding by design or in multivariate models. Findings concerning the association

**Table 3** Least square means and 95% CIs for systolic blood pressure and diastolic blood pressure, by gender of the deceased parent during childhood and further classified according to the cause of death and the age at loss (N=48 624)

| Type of exposure | Model 1*<br>LS mean (95% CI) | P value | Model 2†<br>LS mean (95% CI) | P value | Model 3‡<br>LS mean (95% CI) | P value |
|---|---|---|---|---|---|---|
| **Systolic blood pressure, mm Hg** | | | | | | |
| Unexposed | 126.09 (125.99 to 126.20) | – | 126.12 (125.96 to 126.28) | – | 127.81 (126.65 to 128.98) | – |
| **Death of a mother§** | | | | | | |
| Cause of death | | | | | | |
| Unnatural death | 125.35 (123.05 to 127.66) | 0.53 | 125.57 (123.27 to 127.87) | 0.64 | 127.43 (124.84 to 130.02) | 0.75 |
| Cardiovascular death | 126.42 (123.84 to 129) | 0.81 | 126.30 (123.73 to 128.87) | 0.89 | 127.87 (125 to 130.73) | 0.97 |
| Other natural death | 126.62 (125.72 to 127.51) | 0.25 | 126.47 (125.57 to 127.37) | 0.44 | 128.27 (126.81 to 129.73) | 0.32 |
| Child's age at loss | | | | | | |
| ≤5 years | 124.99 (123.15 to 126.84) | 0.24 | 124.75 (122.91 to 126.59) | 0.14 | 126.55 (124.36 to 128.74) | 0.16 |
| 6–12 years | 126.29 (125.01 to 127.56) | 0.77 | 126.16 (124.89 to 127.44) | 0.95 | 127.79 (126.07 to 129.51) | 0.91 |
| 13–18 years | 126.95 (125.80 to 128.10) | 0.14 | 126.91 (125.76 to 128.07) | 0.18 | 129 (127.37 to 130.63) | 0.05 |
| **Death of a father§** | | | | | | |
| Cause of death | | | | | | |
| Unnatural death | 125.51 (124.52 to 126.51) | 0.25 | 125.56 (124.57 to 126.55) | 0.27 | 127.59 (126.07 to 129.11) | 0.66 |
| Cardiovascular death | 127.02 (126.02 to 128.02) | 0.07 | 127.10 (126.10 to 128.09) | 0.06 | 128.47 (126.95 to 129.99) | 0.20 |
| Other natural death | 126.62 (125.88 to 127.37) | 0.17 | 126.68 (125.93 to 127.42) | 0.15 | 128.33 (126.96 to 129.70) | 0.18 |
| Child's age at loss | | | | | | |
| ≤5 years | 126.60 (125.58 to 127.63) | 0.33 | 126.74 (125.70 to 127.77) | 0.26 | 128.55 (127.01 to 130.09) | 0.21 |
| 6–12 years | 126.31 (125.41 to 127.20) | 0.64 | 126.38 (125.48 to 127.28) | 0.59 | 128.13 (126.66 to 129.59) | 0.59 |
| 13–18 years | 126.56 (125.85 to 127.28) | 0.20 | 126.58 (125.86 to 127.30) | 0.22 | 128.28 (126.92 to 129.63) | 0.27 |
| **Diastolic blood pressure, mm Hg** | | | | | | |
| Unexposed | 72.90 (72.82 to 72.99) | – | 72.93 (72.80 to 73.06) | – | 73.92 (72.98 to 74.86) | – |
| **Death of a mother§** | | | | | | |
| Cause of death | | | | | | |
| Unnatural death | 70.68 (68.84 to 72.51) | 0.02 | 70.67 (68.84 to 72.51) | 0.02 | 72.28 (70.18 to 74.37) | 0.09 |
| Cardiovascular death | 72.09 (70.04 to 74.14) | 0.44 | 72.13 (70.07 to 74.18) | 0.44 | 72.61 (70.29 to 74.92) | 0.22 |
| Other natural death | 72.69 (71.98 to 73.40) | 0.55 | 72.73 (72.01 to 73.45) | 0.58 | 73.81 (72.63 to 74.99) | 0.77 |
| Child's age at loss | | | | | | |
| ≤5 years | 71.83 (70.37 to 73.30) | 0.15 | 71.86 (70.40 to 73.33) | 0.16 | 74.35 (73.12 to 75.58) | 0.08 |
| 6–12 years | 72.16 (71.14 to 73.17) | 0.15 | 72.20 (71.18 to 73.22) | 0.16 | 74.35 (73.25 to 75.46) | 0.17 |
| 13–18 years | 72.72 (71.80 to 73.63) | 0.69 | 72.74 (71.82 to 73.66) | 0.69 | 73.37 (72.14 to 74.60) | 0.81 |
| **Death of a father§** | | | | | | |
| Cause of death | | | | | | |
| Unnatural death | 72.43 (71.64 to 73.22) | 0.24 | 72.36 (71.57 to 73.15) | 0.17 | 74 (72.69 to 75.31) | 0.18 |
| Cardiovascular death | 73.62 (72.82 to 74.41) | 0.08 | 73.59 (72.79 to 74.38) | 0.11 | 72.52 (70.75 to 74.29) | 0.30 |
| Other natural death | 73.44 (72.85 to 74.03) | 0.08 | 73.39 (72.80 to 73.99) | 0.14 | 73.16 (71.77 to 74.55) | 0.17 |
| Child's age at loss | | | | | | |
| ≤5 years | 72.96 (72.14 to 73.78) | 0.90 | 72.84 (72.01 to 73.67) | 0.84 | 73.76 (72.52 to 75.01) | 0.77 |
| 6–12 years | 73.47 (72.76 to 74.19) | 0.12 | 73.39 (72.67 to 74.11) | 0.22 | 74.25 (73.07 to 75.43) | 0.34 |
| 13–18 years | 73.07 (72.50 to 73.64) | 0.58 | 73.07 (72.50 to 73.65) | 0.62 | 73.96 (72.87 to 75.06) | 0.79 |

*Model 1 was unadjusted.
†Model 2 was adjusted for parental socioeconomic status.
‡Model 3 was adjusted for parental socioeconomic status, depression, anxiety, body mass index, cardiorespiratory fitness, number of friends, frequency of talking with friends confidentially, having a confidant, number of cigarettes smoked per day and risky drinking behaviour.
§Only men without missing data on type of exposure were include.
LS, least square.

between other types of childhood adversities—often defined as a cumulative index of events related to abuse, neglect or family dysfunction—and hypertension in middle age or old age are more consistent,[15 38 39] possibly due to the inclusion of a larger number of adverse life events and biases associated with the retrospective assessment of exposure. Our study extends knowledge in this field by focusing on BP assessed at a young age, using

**Table 4** ORs for high blood pressure at conscription according to exposure to death of a parent during childhood (N=48 624)

| Type of exposure | Events/N | Model 1* OR (95% CI) | Model 2† OR (95% CI) | Model 3‡ OR (95% CI) |
|---|---|---|---|---|
| Unexposed | 9166/45 120 | 1 | 1 | 1 |
| Any loss | 776/3504 | 1.12 (1.03 to 1.21) | 1.11 (1.02 to 1.21) | 1.10 (1 to 1.20) |
| Cause of death of the parent§ | | | | |
| Unnatural death | 119/634 | 0.91 (0.74 to 1.11) | 0.91 (0.74 to 1.11) | 0.93 (0.75 to 1.15) |
| Cardiovascular death | 142/606 | 1.20 (0.99 to 1.45) | 1.20 (0.99 to 1.45) | 1.15 (0.95 to 1.40) |
| Other natural cause | 383/1608 | 1.23 (1.09 to 1.38) | 1.21 (1.08 to 1.37) | 1.20 (1.06 to 1.35) |
| Child's age at loss | | | | |
| ≤5 years | 218/1101 | 0.97 (0.83 to 1.13) | 0.96 (0.82 to 1.12) | 0.94 (0.81 to 1.11) |
| 6–12 years | 226/979 | 1.18 (1.01 to 1.37) | 1.16 (1 to 1.36) | 1.15 (0.98 to 1.35) |
| 13–18 years | 332/1424 | 1.19 (1.05 to 1.35) | 1.19 (1.05 to 1.35) | 1.18 (1.03 to 1.34) |
| Gender of deceased parent§ | | | | |
| Mother | 196/876 | 1.13 (0.96 to 1.33) | 1.11 (0.95 to 1.30) | 1.12 (0.94 to 1.32) |
| Cause of mother's death | | | | |
| Unnatural death | 21/99 | 1.06 (0.65 to 1.71) | 1.08 (0.67 to 1.76) | 1.06 (0.64 to 1.77) |
| Cardiovascular death | 14/79 | 0.84 (0.47 to 1.51) | 0.83 (0.46 to 1.48) | 0.88 (0.49 to 1.60) |
| Other natural death | 155/660 | 1.20 (1 to 1.44) | 1.18 (0.98 to 1.41) | 1.18 (0.98 to 1.42) |
| Child's age at mother's death | | | | |
| ≤5 years | 25/155 | 0.75 (0.49 to 1.16) | 0.73 (0.47 to 1.12) | 0.65 (0.41 to 1.04) |
| 6–12 years | 81/323 | 1.31 (1.02 to 1.69) | 1.29 (1 to 1.66) | 1.27 (0.97 to 1.65) |
| 13–18 years | 90/398 | 1.15 (0.91 to 1.45) | 1.14 (0.90 to 1.44) | 1.19 (0.93 to 1.52) |
| Father | 493/2180 | 1.15 (1.03 to 1.27) | 1.15 (1.03 to 1.28) | 1.13 (1.01 to 1.26) |
| Cause of father's death | | | | |
| Unnatural death | 98/535 | 0.88 (0.71 to 1.10) | 0.88 (0.70 to 1.10) | 0.91 (0.72 to 1.14) |
| Cardiovascular death | 128/527 | 1.26 (1.03 to 1.54) | 1.26 (1.03 to 1.54) | 1.19 (0.97 to 1.47) |
| Other natural death | 228/948 | 1.24 (1.07 to 1.44) | 1.24 (1.06 to 1.45) | 1.21 (1.03 to 1.42) |
| Child's age at father's death | | | | |
| ≤5 years | 106/498 | 1.06 (0.85 to 1.32) | 1.06 (0.85 to 1.33) | 1.06 (0.84 to 1.33) |
| 6–12 years | 145/656 | 1.11 (0.92 to 1.34) | 1.11 (0.92 to 1.34) | 1.10 (0.90 to 1.34) |
| 13–18 years | 242/1026 | 1.21 (1.05 to 1.40) | 1.21 (1.05 to 1.40) | 1.17 (1.01 to 1.37) |
| Child's age at loss§ | | | | |
| ≤5 years | | | | |
| Cause of parent's death | | | | |
| Unnatural death | 17/126 | 0.61 (0.37 to 1.02) | 0.61 (0.36 to 1.02) | 0.61 (0.36 to 1.05) |
| Cardiovascular death | 12/61 | 0.96 (0.51 to 1.81) | 0.97 (0.52 to 1.84) | 1.01 (0.53 to 1.93) |
| Other natural death | 57/258 | 1.11 (0.83 to 1.49) | 1.09 (0.81 to 1.47) | 1.04 (0.76 to 1.41) |
| 6–12 years | | | | |
| Cause of parent's death | | | | |
| Unnatural death | 41/229 | 0.86 (0.61 to 1.20) | 0.84 (0.60 to 1.19) | 0.85 (0.59 to 1.21) |
| Cardiovascular death | 46/185 | 1.30 (0.93 to 1.81) | 1.29 (0.92 to 1.81) | 1.24 (0.88 to 1.77) |
| Other natural death | 139/565 | 1.28 (1.06 to 1.55) | 1.26 (1.04 to 1.53) | 1.24 (1.01 to 1.52) |
| 13–18 years | | | | |
| Cause of parent's death | | | | |
| Unnatural death | 61/279 | 1.10 (0.83 to 1.46) | 1.11 (0.84 to 1.48) | 1.15 (0.86 to 1.54) |
| Cardiovascular death | 84/360 | 1.19 (0.93 to 1.53) | 1.19 (0.93 to 1.52) | 1.12 (0.87 to 1.45) |
| Other natural death | 187/785 | 1.23 (1.04 to 1.45) | 1.22 (1.03 to 1.44) | 1.21 (1.02 to 1.44) |

*Model 1 was unadjusted.
†Model 2 was adjusted for parental socioeconomic status.
‡Model 3 was adjusted for parental socioeconomic status, depression, anxiety, body mass index, cardiorespiratory fitness, number of friends, frequency of talking with friends confidentially, having a confidant, number of cigarettes smoked per day and risky drinking behaviour.
§Only men with no missing data on this type of exposure were included.

prospectively recorded information on parent's death from a high-quality nationwide register, rather than relying on retrospectively collected, questionnaire-based information on exposure and analysing a very large sample that allowed us to conduct subanalyses to test hypotheses that may be supportive of a causal effect, if any.

Since an important challenge in studies regarding the association between bereavement and CVDs is related to the separation of the stress-related effect from confounding by genetic and environmental cardiovascular risk factors shared by family members, we performed analyses by the parents' cause of death classified as cardiovascular, other natural and unnatural deaths. Furthermore, to explore dose-response patterns and potential sensitive periods, we performed analyses according to the type of deceased parent or age at loss. We found no association between parental death during childhood and our continuous BP measures neither in case of overall exposure, nor when exposure was categorised by the parents' cause of death or gender or the child's age at loss. These results are indicative of a lack of a causal effect.

The finding that a modestly increased hypertension risk was observed only in case of parental deaths due to natural causes but not in case of parental deaths due to unnatural causes—which are often associated with more severe stress[6 16 23]—suggests that residual confounding by familial cardiovascular risk factors is a likely explanation for the association between parental death during childhood and hypertension. A family history of metabolic disorders and early CVD is a well-established risk factor of hypertension in children and youth.[56] An alternative explanation may be that severe stress increases the risk of hypertension in youth only among individuals with a genetic susceptibility to cardiometabolic disorders; children exposed to parental death due to natural causes are more likely to have such a susceptibility than children who lost a parent due to unnatural causes. A further though—in light of the fact that exposure in young age was not associated with the outcome—less likely explanation for the association between parental death due to natural causes and hypertension is that natural deaths may be proceeded by a long period of disease which may induce chronic stress for family members.[57] The stronger association between parental death in adolescence and hypertension than in earlier periods of life—besides better statistical power—is also likely to be due to residual confounding, given the higher proportion of parental deaths related to cardiometabolic conditions in the older age group.

A possible explanation for the limited evidence for an association between the death of a parent and BP in our study, in contrast to the large body of evidence documenting a link between psychological stress and hypertension in adulthood[58] and childhood adversities and hypertension in middle age or old age[15 38 39] may be related to the age of the BP assessment. Though children who lost a parent may experience chronic stress, the accumulation

of further childhood (eg, adverse socioeconomic circumstances, low social support, abuse or neglect, poor mental health) or adult adversities (eg, low educational attainment, difficulties in attaching to a partner, difficulties on the labour marker) and the subsequent allostatic load may need to act for a longer period than what we studied to induce abnormal BP changes.[59] At the age of 18–20 years differences in BP may still be small across individuals.[43] Allostatic load may need to interact with the age-related vulnerability to increase BP later in life. This hypotheses is supported by findings of Su and associates who assessed BP on average 13 times between the age of 5 and 38 years, and found that mean SBP and DBP were similar in the first two decades of life among groups exposed and unexposed to adverse childhood experiences, but from the third decade of life differences among exposure groups in levels of BP became evident and with time increasingly important; the increase in mean BP was steeper with a higher number of adverse childhood experiences.[43]

Our findings need to be considered also in light of our study's limitations. First, since only one or two BP measurements were taken at conscription, we did not have the possibility to consider intra-individual variations in BP,[60] neither to reduce an eventual bias related to white-coat hypertension. Nevertheless, several measurements or ambulatory BP monitoring[19 61] may not be feasible in large epidemiological studies such as ours, as these measures may limit the sample size and may generate selection bias, for example, children with parental death and/or hypertension may be more or less likely to participate in such a study compared with others. The potential misclassification introduced by the single measurement in our study is likely to be non-differential and may eventually result in an underestimation of the true effect. Nevertheless, the BP measurement at conscription followed the standard measurement for screening for hypertension in primary care. Though one measurement of BP is not sufficient to confirm a clinical diagnosis of hypertension,[62] it has a screening value as the BP measure in this cohort has been reported to be an important predictor of later CVD and mortality.[46 63] Future studies with repeated measurements of BP ranging from childhood to later adulthood are needed to investigate the link between parental death and BP. Second, since the conscription cohort included only men, it is not clear whether the results can be generalised to women. Though women may be more sensitive to psychological stress with respect to hypertension than men,[64] earlier studies in this field investigating the association between adverse childhood experiences and youth BP have mixed results in term of gender differences.[43 65 66] Similarly, it is not clear to what extent our findings may be generalisable to current Swedish young adults or to populations with different risk factor profiles, support for bereaved children and healthcare.[67] Third, since information on most of our covariates was available only from conscription, we do not know whether they are confounders or mediators of the investigated associations. Though theoretical considerations and our

measurement post-exposure favour regarding them as mediators, we cannot exclude the possibility that some of these measures are indicators of familial characteristics and may thus be proxies for confounding by factors that cluster in families.[17] Nevertheless, adjustment for factors in model 3 did not substantially affect our estimates; these covariates were generally weakly associated with hypertension. The finding that natural deaths but not unnatural deaths, which are less likely to be affected by familial cardiovascular risk, were associated with an increased hypertension risk and that associations were unchanged after adjusting for these suggests the presence of confounding by factors that we did not consider; potential candidates include genetics, further socioeconomic factors, housing, diet and familial cardiometabolic disorders. Fourth, though our sample was very large, the power in some of our subanalyses, for example, when categorising exposure according to the deceased parents' gender, cause of death and the child's age at loss and the risk of hypertension, may have been low to detect a modest effect at the conventional p<0.05 level.

We found no evidence that death of a parent was associated with continuous SBP and DBP measured at the age of 18–20. The fact that the association between loss of a parent and the modest risk of hypertension was observed in case of parental deaths due to natural causes but not in case of unnatural deaths, may be indicative of residual confounding by genetic and environmental factors shared by family members.

### Author affiliations
$^1$Department of Global Public Health, Karolinska Institutet, Stockholm, Sweden
$^2$Department of Public Health Sciences, Stockholm University, Stockholm, Sweden
$^3$Institute of Environmental Medicine, Karolinska Institutet, Stockholm, Sweden
$^4$Department of Public Health and General Practice, Faculty of Medicine, Norwegian University of Science and Technology, Trondheim, Norway
$^5$Centre for Health Equity Studies, Stockholm University and Karolinska Institutet, Stockholm, Sweden
$^6$Centre for Epidemiology and Community Medicine, Stockholm County Council, Stockholm, Sweden
$^7$Statistical Office, Capital Institute of Pediatrics, Beijing, China

**Contributors** Conceptualisation: HC, KDL and IJ. Data curation: HC, TH and KDL. Formal analysis: HC. Funding acquisition: KDL and HC. Methodology: HC, KDL, TH, IJ, MR, YF, LM and YL. Resources: TH and KDL. Supervision: KDL, TH, MR, and YF. Writing—original draft: HC. Writing—review and editing: HC, KDL, TH, IJ, MR, YF, YL and LM.

**Funding** The study was supported by the Swedish Council for Working Life and Social Research (grant number: 2015-00837), by Karolinska Institutet's Research Foundation (grant number: 2018-01924) and by the China Scholarship Council (grant number: 201700260296).

**Competing interests** None declared.

**Patient consent for publication** Not required.

**Ethics approval** The study was approved by the Regional Ethics Board in Stockholm (reference numbers: 2004-693/5, 2008/323-32 and 2010/604-32).

**Provenance and peer review** Not commissioned; externally peer reviewed.

**Data availability statement** Data may be obtained from a third party and are not publicly available.

**ORCID iDs**
Hua Chen http://orcid.org/0000-0002-4884-3360
Krisztina D. László http://orcid.org/0000-0002-4695-477X

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
