## [Reviewer comments · BMJ Open]

ARTICLE DETAILS

TITLE (PROVISIONAL)	Death of a parent during childhood and blood pressure in youth: a population-based cohort study of Swedish men
AUTHORS	Chen, Hua; Hemmingsson, Tomas; Janszky, Imre; Rostila, Mikael; Forsell, Yvonne; Meng, Linghui; Liang, Yajun; Laszlo, Krisztina

VERSION 1 – REVIEW

REVIEWER	Andrew H. Tran Nationwide Children's Hospital, Columbus, OH, USA
REVIEW RETURNED	15-Sep-2020

GENERAL COMMENTS	Overview: Past research has shown that childhood adversity is associated with increased risk for hypertension in middle aged and older adults. However, the data linking childhood adversity and younger populations is mixed. The authors of this current study hypothesized that parental death during childhood would be associated with hypertension during early adulthood in men. Overall, there was no significant difference in systolic and diastolic blood pressure between subjects who lost their parents and those who did not. This was a large, well-written study that examined multiple factors regarding parental death and hypertension. Minor Points: 1. Please clarify the research ethics approval or ethics exemption for the study in the methods section.2. Methods: Page 7, line 40 – Please clarify the assigned work capacity measures to the stanine scoring.3. The authors discuss on page 12, lines 49-58, that the increased risk of hypertension in parental deaths due to natural causes (seen in Table 4, line 24) could potentially be explained by residual confounding for familial cardiovascular factors. It is odd then that there is not a significantly increased risk of hypertension for parental cardiovascular death in the models on Table 4, line 22. It seems that parental cardiovascular death should have the most risk for offspring hypertension. Please comment on this.4. Please comment further regarding potential differences in risk of hypertension in children regarding gender of parent as it seems from Table 4, line 59 that there may be increased risk of hypertension in children with the death of their father (Table 4, line 59).5. Table 4, line 12 – The word “mother’s” should be “father’s.”
---

REVIEWER	Jan Terock University Medicine Greifswald, Dept. of Psychiatry and Psychotherapy
REVIEW RETURNED	08-Nov-2020

GENERAL COMMENTS	General comment: The authors present results from a huge (N>48,000) Swedish sample of male military recruits. The study applied a prospective design to investigate whether the potentially traumatic event of loss of a parent during childhood was associated with blood pressure at young age of 18-20 yrs. Although the authors performed different subgroup analyses and included various potential confounders, no significant associations could be observed. A major weakness of this study is the small age range and the very young age of participants, an age, where CVDs are typically not yet fully developed. The restriction to only male participants is another important weakness. However, it is a strength at the same time as there is very limited information on this particular age group available as yet. It would also have been interesting to know more about other childhood adversities. Strengths include the enormous sample size and the very detailed characterization of the sample. Also, the statistical procedure is sound and the paper is well written. In all, I feel the manuscript is a relevant contribution to the field. Introduction: The introduction provides an excellent review of the current literature on (a) the association between childhood trauma and BP as well as between (b) high BP in youth and later health conditions. However, no information is given about underlying biological mechanisms, which may link childhood trauma and high BP. In particular, there is broad evidence on neuroendocrine alterations including the HPA-axis as well as the RAAS which may link these two factors. In general, a passage on a pathophysiological working model is necessary to justify the analyses and to explain why one would expect an association between child trauma and high BP at all. Also, is there evidence from previous studies on the association of parental loss and other CVDs, particularly in the group of youths? The authors performed additional analyses on differential effects caused by the parent's cause of death, the gender of the deceased parent and the child's age at loss. However, it remains unclear, why the authors picked these variables. Is there evidence from previous studies suggesting an effect? The introduction needs to be elaborated regarding previous evidence. Various potential confounders were taken into consideration in the fully adjusted analyses. These should be briefly introduced, e. g. may mentioning that differences in the confounder set may have contributed to the inconsistencies in previous studies. The hypotheses should be described more clearly. What exactly did the authors expect (also regarding the role of the confounders). Methods: What was the reason behind differentiating parent's death into "unnatural", "non-rheumatic CVD" and "other natural causes"? The selection appears random. Why not include other causes of death? Why did the authors analyse non-rheumatoid CVD separately? It does not really become clear when a childhood experience was classified as "adverse". Was there a scoring procedure e. g. for the paternal alcohol use? When was a severe illness of a parent classified as adversity? Was it more severe, when both parents suffered from a severe illness? This needs to be explained in more detail. Also, are there references for such a procedure? Results:
--

	The presentation of the results is clear and concise. I have no concerns regarding this section. Discussion: The paragraph beginning on page 11 “An important challenge...” finally provides an explanation for the separate analyses of deaths from CVD. I think this is an important passage, which would better fit in the introduction section. Also, the following passage now explain why the authors expected that the gender of the parent’s death as well as the age of the participants may have played a role. All these considerations need to be presented in the introduction. In general, the authors did a good job in discussing the association between natural death of a parent and hypertension. Still, the interesting finding that results hardly differed between the unadjusted and the fully adjusted analyses is largely neglected in the discussion. The finding of nearly no influence of important confounders should explicitly be acknowledged.
--	--

VERSION 1 – AUTHOR RESPONSE

Reviewer #1:

Dr. Andrew Tran, Nationwide Children's Hospital

Comment:

Overview:

Past research has shown that childhood adversity is associated with increased risk for hypertension in middle aged and older adults. However, the data linking childhood adversity and younger populations is mixed. The authors of this current study hypothesized that parental death during childhood would be associated with hypertension during early adulthood in men. Overall, there was no significant difference in systolic and diastolic blood pressure between subjects who lost their parents and those who did not. This was a large, well-written study that examined multiple factors regarding parental death and hypertension.

Response:

Thank you very much for your positive evaluation.

Comment:

Minor Points:

- 1. Please clarify the research ethics approval or ethics exemption for the study in the methods section.*

Response:

We have now added the number of the ethical permit in the Method section; please see page 6, paragraph 4.

Comment:

2. *Methods: Page 7, line 40 – Please clarify the assigned work capacity measures to the stanine scoring.*

Response:

We have now clarified the work capacity measures that we assigned to the stanine scoring (page 8, paragraph 2).

Comment:

3. *The authors discuss on page 12, lines 49-58, that the increased risk of hypertension in parental deaths due to natural causes (seen in Table 4, line 24) could potentially be explained by residual confounding for familial cardiovascular factors. It is odd then that there is not a significantly increased risk of hypertension for parental cardiovascular death in the models on Table 4, line 22. It seems that parental cardiovascular death should have the most risk for offspring hypertension. Please comment on this.*

Response:

Based on suggestions from reviewer 2, we now present in more detail in the Introduction our hypotheses regarding the role of the deceased parent's gender, cause of death and the child's age at loss in the association between death of a parent and CVD (page 5, paragraph 2).

In line with what you highlighted, we expected an association between the death of a parent in childhood and the risk of hypertension due to (1) adverse stress-related psychosocial, lifestyle-related or physiological changes and/or (2) confounding by cardiovascular risk factors shared by family members. The second potential explanation would be more important in case of deaths due to cardiovascular diseases than other natural deaths but is likely to contribute also to the explanation of the association between other natural deaths and the risk of hypertension. The natural causes other than cardiovascular diseases in our study included a wide range of diseases; many of them (e.g., cancer and endocrine and metabolic diseases) have common risk factors with cardiovascular diseases (See e.g., Peters R, et al. Common risk factors for major noncommunicable disease, a systematic overview of reviews and commentary: the implied potential for targeted risk reduction. *Ther Adv Chronic Dis.* 2019; 10:2040622319880392.).

The point estimates for the risk of hypertension in case of death due to cardiovascular and other natural causes were very similar and the corresponding confidence intervals largely overlapped (1.15 (0.95-1.40) versus 1.20 (1.06-1.35) in the fully adjusted models. The associations in case of parental deaths due to cardiovascular diseases and hypertension did not reach statistical significance at the conventional $p=0.05$ level, potentially due to the lower number of participants with hypertension exposed to such losses (142/606 events as compared to 383/1608 events for losses due to other natural deaths). We have added the number of events/N corresponding to each exposure group to Table 4 to highlight this difference in statistical power. We acknowledge among the limitations of our study that

though our sample size was very large, in some of the sub-analyses we may have had limited statistical power to detect modest associations at the conventional $p < 0.05$ statistical level (page 16, paragraph 1).

Comment:

4. Please comment further regarding potential differences in risk of hypertension in children regarding gender of parent as it seems from Table 4, line 59 that there may be increased risk of hypertension in children with the death of their father (Table 4, line 59).

Response:

We recognize that opinions differ with respect to the issue of statistical significance. We agree with the experts who strongly discourage the use p-value in a dichotomous way where the results are reported as statistically significant or non-significant findings; see for example Amrhein et al. Retire statistical significance. Nature 2019 567:305-7, the recent statement of the American Statistical Association on this issue (Wasserstein RL, Lazar NA. The ASA's statement on p-values: context, process and purpose. Am Stat. 2016;70(2):129–133), but also the latest Vancouver recommendations (Recommendations for the Conduct, Reporting, Editing, and Publication of Scholarly Work in Medical Journals. Updated December 2019, 2019. <http://www.icmje.org/icmje-recommendations.pdf>). Thus, we presented point estimates and confidence intervals.

The odds ratios and the 95% confidence intervals corresponding to the association between the death of the mother and father and the risk of hypertension are 1.12 (0.94-1.32) and 1.13 (1.01-1.26), respectively. The point estimates are almost identical, and the confidence intervals largely overlap. Given the substantially higher number of paternal than maternal deaths during the study period, our interpretation is rather that power to detect a statistically significant association at the conventional $p = 0.05$ level was limited in case of maternal losses and not that there was evidence for a substantial difference in the association between bereavement and the risk of hypertension according to the gender of the deceased parent. We add the number of outcomes and each exposure category in Table 4 to highlight this difference in statistical power between analyses concerning maternal and paternal deaths. As mentioned, we acknowledge among the limitations of our study that though our sample size was very large, in some of the sub-analyses concerning subtype of bereavement and the risk of hypertension, we may have had limited statistical power to detect modest associations at the conventional $p < 0.05$ statistical level (page 16, paragraph 1).

Comment:

5. Table 4, line 12 – The word “mother’s” should be “father’s.”

Response:

Thank you, we have rectified the text according to your suggestion.

Reviewer# 2:

Dr. Jan Terock, HELIOS Hansekllinikum Stralsund

Comment:

General comment:

The authors present results from a huge (N>48,000) Swedish sample of male military recruits. The study applied a prospective design to investigate whether the potentially traumatic event of loss of a parent during childhood was associated with blood pressure at young age of 18-20 yrs. Although the authors performed different subgroup analyses and included various potential confounders, no significant associations could be observed.

A major weakness of this study is the small age range and the very young age of participants, an age, where CVDs are typically not yet fully developed. The restriction to only male participants is another important weakness. However, it is a strength at the same time as there is very limited information on this particular age group available as yet. It would also have been interesting to know more about other childhood adversities.

Strengths include the enormous sample size and the very detailed characterization of the sample. Also, the statistical procedure is sound and the paper is well written. In all, I feel the manuscript is a relevant contribution to the field.

Response:

Thank you very much for your positive review on our manuscript.

We now mention among the limitations of the study that we only had measures of blood pressure on one or two occasions at the age of 18-20 years and that studies with repeated blood pressure measures ranging from childhood to adulthood are needed to further investigate the question we addressed (page 15, paragraph 1). We have clarified the rationale for our choice of the measures of adverse childhood experiences, how they were measured and categorized and how we calculated the cumulative index of adverse childhood experiences (page 9, paragraph 1).

Comment:

Introduction:

The introduction provides an excellent review of the current literature on (a) the association between childhood trauma and BP as well as between (b) high BP in youth and later health conditions. However, no information is given about underlying biological mechanisms, which may link childhood trauma and high BP. In particular, there is broad evidence on neuroendocrine alterations including the HPA-axis as well as the RAAS which may link these two factors. In general, a passage on a pathophysiological

working model is necessary to justify the analyses and to explain why one would expect an association between child trauma and high BP at all.

Response:

We now describe in more detail the underlying biological mechanisms that may link childhood trauma to hypertension; please see page 4 (paragraph 1).

Comment:

Also, is there evidence from previous studies on the association of parental loss and other CVDs, particularly in the group of youths?

Response:

Knowledge regarding the association between parental death during childhood and other CVDs than hypertension is very limited. To our knowledge, only three studies analyzed the relation between parental death during childhood and CVD; one focused on cardiovascular mortality in young age (Li et al., 2014), one focused on cardiovascular mortality in old age (Smith et al., 2014), while a third focused on ischemic heart diseases and stroke up to middle age (Chen et al., 2020). We now cite these studies in our Introduction (page 4, paragraph 1).

Comment:

The authors performed additional analyses on differential effects caused by the parent's cause of death, the gender of the deceased parent and the child's age at loss. However, it remains unclear, why the authors picked these variables. Is there evidence from previous studies suggesting an effect? The introduction needs to be elaborated regarding previous evidence.

Various potential confounders were taken into consideration in the fully adjusted analyses. These should be briefly introduced, e. g. may mentioning that differences in the confounder set may have contributed to the inconsistencies in previous studies.

The hypotheses should be described more clearly. What exactly did the authors expect (also regarding the role of the confounders).

Response:

We now clarify in the Introduction our hypothesis regarding the role of the parent's gender, cause of death and the child's age at loss in the association between death of a parent and later health (page 5, paragraph 2). We also discuss about the importance of confounding by familial risk factors in these associations. We mention that some of the differences in the findings of the earlier studies in this area may have been due to different considerations about confounding (page 4, paragraph 2).

Comment:

Methods:

What was the reason behind differentiating parent's death into "unnatural", "non-rheumatic CVD" and "other natural causes"? The selection appears random. Why not include other causes of death? Why did the authors analyse non-rheumatoid CVD separately?

Response:

We now present our hypotheses concerning the role of the parent's cause of death in the association between death of a parent and later health in the Introduction (page 5, paragraph 2). The effect of exposure to death of a parent due to natural causes, in particular those due to CVD, on the risk of hypertension is likely to be confounded by genetic and environmental cardiovascular factors shared by family members, e.g., socioeconomic factors, lifestyle, mental health, cardiometabolic risk factors and morbidity etc.; in contrast, the importance of familial confounding by cardiovascular risk factors is likely to be substantially more modest in case of unnatural deaths (Li, et al., 2014; Rostila, et al, 2017).

We did not include rheumatic heart diseases in the overall cardiovascular death of the parent category since they have a different etiology than hypertension and most other CVDs. They are primarily due to an untreated streptococcal infection, are generally preventable by administering antibiotics in case of streptococcal infection, are a disease of poverty and, though it can occur in any ages, they are most common in children. Thus, as they were extremely rare among parents in Sweden during the study period (n=34), we believe the choice of whether to include rheumatic heart diseases to cardiovascular or other natural deaths would not influence our results.

Comment:

It does not really become clear when a childhood experience was classified as "adverse". Was there a scoring procedure e. g. for the paternal alcohol use? When was a severe illness of a parent classified as adversity? Was it more severe, when both parents suffered from a severe illness? This needs to be explained in more detail. Also, are there references for such a procedure?

Response:

We now present that our choice of the investigated adverse childhood experiences was based on previous literature and the measures that were available in the questionnaire completed at conscription. The most commonly used measures of adverse childhood experiences in the literature are related to abuse, neglect and family dysfunction. There is increasing evidence for a cumulative effect of adverse childhood experiences and the risk of CVD later in life. Often, due to restraints of using large data and with information collected decades ago, the number of measures is summed up to calculate a cumulative index of childhood adversity. We refer to studies and reviews using or describing this approach widely used in the literature. We have now clarified how each investigated measure of childhood adversity was categorized into a binary variable; we also mention that we calculated our cumulative index of adverse childhood experiences other than the death of a parent by summing the

number of assessed adverse childhood experiences that the study participants were exposed to (page 9, paragraph 1).

Comment:

Results:

The presentation of the results is clear and concise. I have no concerns regarding this section.

Response:

Thank you.

Comment:

Discussion:

The paragraph beginning on page 11 “An important challenge...” finally provides an explanation for the separate analyses of deaths from CVD. I think this is an important passage, which would better fit in the introduction section.

Also, the following passage now explain why the authors expected that the gender of the parent’s death as well as the age of the participants may have played a role. All these considerations need to be presented in the introduction.

Response:

We now describe in the Introduction our hypotheses concerning the role of the parents’ cause of death, gender and the child’s age at loss and have somewhat modified the Discussion to prevent repetition with the content of the Introduction. Please see page 5, paragraph 2 and page 13, paragraph 1.

Comment:

In general, the authors did a good job in discussing the association between natural death of a parent and hypertension. Still, the interesting finding that results hardly differed between the unadjusted and the fully adjusted analyses is largely neglected in the discussion. The finding of nearly no influence of important confounders should explicitly be acknowledged.

Response:

We now discuss on page 15 that for most of the covariates we adjusted for we had information only from conscription, thus we do not know whether they were confounders or mediators. Though theoretical considerations and our measurement after exposure could favor regarding them as mediators, we cannot exclude the possibility that some of these measures are indicators of characteristics of the family (Chen et al., 2020). The finding that natural deaths but not unnatural deaths, which are less likely to be affected by cardiovascular risk factors in families, were associated with an

increased hypertension risk and that associations were unchanged after adjusting for these suggests the presence of residual confounding; potential candidates include genetics, additional socioeconomic factors, housing, diet and cardiometabolic disorders in the family. We have earlier presented the association between exposure and covariates in Table 1. We now include also a table with the associations between the covariates and our outcomes (Supplementary table 1). The association between covariates and high blood pressure was generally modest, potentially further contributing to the small impact of the multivariate adjustments on our estimates.

VERSION 2 – REVIEW

REVIEWER	Tran, Andrew Nationwide Children's Hospital
REVIEW RETURNED	10-Mar-2021

GENERAL COMMENTS	I have no further comments at this time. The authors have addressed my previous comments. Thank you.
--

REVIEWER	Terock, Jan HELIOS Hanselinikum Stralsund, Psychiatry and Psychotherapy
REVIEW RETURNED	24-Feb-2021

GENERAL COMMENTS	The authors present a thorough revision. I have no further comments and recommend acceptance.
---